# A Semi-Analytical Method for Channel and Pipe Flows for the Linear Phan-Thien-Tanner Fluid Model with a Solvent Contribution

**DOI:** 10.3390/polym14214675

**Published:** 2022-11-02

**Authors:** Matheus Tozo de Araujo, Laison Furlan, Analice Brandi, Leandro Souza

**Affiliations:** 1Department of Applied Mathematics and Statistics, University of Sao Paulo, Sao Carlos 13566-590, Brazil; 2Department of Mathematics and Computer Science, Sao Paulo State University, Presidente Prudente 19060-900, Brazil

**Keywords:** Phan-Thien-Tanner constitutive equation, semi-analytical method, solvent viscosity contribution, pipe flow, channel flow

## Abstract

This work presents a semi-analytical method for laminar steady-state channel and pipe flows of viscoelastic fluids using the Linear Phan-Thien-Tanner (LPTT) constitutive equation, with solvent viscosity contribution. For the semi-analytical method validation, it compares its results and two analytical solutions: the Oldroyd-B model and the simplified LPTT model (without solvent viscosity contribution). The results adopted different values of the dimensionless parameters, showing their influence on the viscoelastic fluid flow. The results include the distribution of the streamwise velocity component and the extra-stress tensor components in the wall-normal direction. In order to investigate the proposed semi-analytical method, different solutions were obtained, both for channel and pipe flows, considering different values of Reynolds number, solvent viscosity contribution in the homogeneous mixture, elongational parameter, shear parameter, and Weissenberg number. The results show that the proposed semi-analytical method can find a laminar solution using the non-Newtonian LPTT model with solvent viscosity contribution and verify the effect of the parameters in the resulting flow field.

## 1. Introduction

Due to the use of viscoelastic fluids in some industries, there is interest in obtaining an analytical solution of constitutive models that describe the behaviour of this type of fluid flow. Several researchers have investigated the analytical solutions of many constitutive models, such as the Oldroyd-B, Giesekus, FENE, and PTT models. Investigations of non-Newtonian fluid flow with heat transfer is also an interesting phenomena [1,2]. Hulsen [3] presented the analytical solution of the Leonov and Giesekus models and some properties such as tensor restrictions and the possibility of arising instabilities due to numerical approximation errors.

An exact solution for tube and slit flows of a FENE-P fluid was found by [4]. Yoo and Choi [5] and Schleiniger and Weinacht [6] present solutions for pipe and channel flows of the Giesekus model for Poiseuille flow. With the same model, Raisi et al. [7] obtained the solution for the Couette-Poiseuille flow. Hayat et al. [8] derived the exact solution for the Oldroyd-B model applied to five different flow problems, and Hayat et al. [9] presented the exact solution of this same model to six different problems of unsteady flow.

More recently, Tomé et al. [10] presented a way to obtain the analytical solution for the Giesekus model (based on [6]), with the pressure gradient being calculated numerically and considering β=0. Furlan et al. [11] studied an analytical solution of the Giesekus model without restrictions on the model parameters.

There are several studies in the literature in which the analytical solution for the LPTT model is obtained. For the simplified LPTT model, the solutions are presented in [12,13,14,15] and the solution for the LPTT model for purely polymeric fluid flow without simplifications is presented in Alves et al. [16]. There is a simplification of the LPTT model equations in the solutions referenced above: the parameter ξ=0 in the LPTT model or the solvent contribution is considered zero in the homogeneous mixture.

The present work shows a semi-analytical method to obtain the flow variables when using the LPTT viscoelastic fluid model for channel and pipe flow without simplifications and considering a solvent contribution in the homogeneous mixture. Channel flow is referred to as the two-dimensional flow between two parallel plates. The proposed method is valid for laminar flow.

The paper is organized as follows. Section 2 presents the governing equations and the mathematical manipulations needed to obtain the semi-analytical method for the LPTT fluid flow with a solvent contribution in the homogeneous mixture; the results obtained using the method proposed are presented in Section 3. The main conclusions are presented in Section 4.

## 2. Mathematical Formulation

The Phan-Thien-Tanner (PTT) constitutive equation was derived from Phan-Thien and Tanner [17] work. The viscoelastic fluid model considered in this analysis is governed by its dimensional form given by:(1)$$f\left(tr(T)\right)T+\lambda \left(\frac{\partial T}{\partial t}+\nabla \cdot \left(\mathrm{uT}\right)-\left(\nabla u-\xi D\right)\cdot T-T\cdot {\left(\nabla u-\xi D\right)}^{T}\right)=2\eta_{p}D,$$
where u denotes the velocity field, *t* is the time, T and D are the extra-stress and deformation-rate tensors, respectively, λ is the fluid relaxation time, $\eta_{p}$ is the polymer viscosity, and ξ is a positive parameter of the PTT model connected with the shear stress behaviour of the fluid.

The function $f\left(tr(T)\right)$ depends on the trace of extra-stress tensor T and determines the form of the PTT model [18]:(i)Linear: $f\left(tr\left(T\right)\right)=1+\frac{\lambda \epsilon }{\eta_{p}}tr\left(T\right)$,(ii)Quadratic: $f\left(tr\left(T\right)\right)=1+\frac{\lambda \epsilon }{\eta_{p}}tr\left(T\right)+\frac{1}{2}{\left(\frac{\lambda \epsilon }{\eta_{p}}tr\left(T\right)\right)}^{2}$,(iii)Exponential: $f\left(tr\left(T\right)\right)=\exp\left(\frac{\lambda \epsilon }{\eta_{p}}tr\left(T\right)\right)$.

The linear form was the original form proposed by Phan-Thien and Tanner [17], and it was used for the PTT model in this work, also called the LPTT model. The parameter ϵ in the function f(tr(T)) is related to the elongational behaviour of the fluid, precluding an infinite elongational viscosity in a simple stretching flow as it would occur for an upper-convected Maxwell model (UCM), in which ϵ=0 [13].

It is considered a fully developed flow for two-dimensional channel and axisymmetric pipe flow (Figure 1). The flow is considered incompressible and isothermal, without the influences of external forces. Furthermore, it is used a compact notation [16,19] with index j=0 for channel flow or j=1 for pipe flow.

In the fully developed flows analyzed here (parallel flow), the velocity and extra-stress tensor components are function of the cross-stream coordinate (*y* or *r*), the pressure gradient in the streamwise direction $P_{x}$ is constant and the continuity equation implies a zero transverse velocity component (v=0).

With the adopted formulation, the *x*-momentum equation does not change with the non-Newtonian constitutive model and can be integrated to give
(2)$$-P_{x}\frac{y}{2^{j}}+\frac{\beta }{Re}\frac{du(y)}{dy}+T_{xy}\left(y\right)=0.$$

The constitutive equation for each extra-stress tensor component, giving the assumption of parallel flow, can be simplified to the following set of expressions
(3)$$\left(1+\frac{\epsilon ReWi}{\left(1-\beta \right)}\left(T_{xx}\left(y\right)+T_{yy}\left(y\right)\right)\right)T_{xx}\left(y\right)=2Wi\left(1-\frac{\xi }{2}\right)T_{xy}\left(y\right)\frac{du(y)}{dy},$$
(4)$$\begin{array}{c} \left(1+\frac{\epsilon ReWi}{\left(1-\beta \right)}\left(T_{xx}\left(y\right)+T_{yy}\left(y\right)\right)\right)T_{xy}\left(y\right)=\frac{\left(1-\beta \right)}{Re}\frac{du(y)}{dy}+\\ -Wi\left(\frac{\xi }{2}T_{xx}\left(y\right)\frac{du(y)}{dy}-\left(1-\frac{\xi }{2}\right)T_{yy}\left(y\right)\frac{du(y)}{dy}\right), \end{array}$$
(5)$$\left(1+\frac{\epsilon ReWi}{\left(1-\beta \right)}\left(T_{xx}\left(y\right)+T_{yy}\left(y\right)\right)\right)T_{yy}\left(y\right)=-Wi\xi T_{xy}\left(y\right)\frac{du(y)}{dy}.$$

The system of Equations (Equation 2)–(Equation 5) is in dimensionless form where the dimensionless parameters $Re=\frac{\rho UL}{\eta_{0}}$ and $Wi=\frac{\lambda U}{L}$ denote the Reynolds and Weissenberg numbers, respectively. The Reynolds number is based in total shear viscosity $\eta_{0}$, and $\eta_{0}=\eta_{s}+\eta_{p}$ where $\eta_{s}$ and $\eta_{p}$ represent the Newtonian solvent and polymeric viscosities, respectively, and ρ is the fluid density, *U* is the velocity scale and *L* is the channel (or pipe) half-width. The amount of Newtonian solvent is controlled by the dimensionless solvent viscosity coefficient, parameter $\beta =\frac{\eta_{s}}{\eta_{0}}$. In Weissenberg number the λ parameter is the relaxation-time of the fluid.

Dividing Equation (Equation 3) by Equation (Equation 5), the relation between the extra-stress tensor components $T_{xx}\left(y\right)$ and $T_{yy}\left(y\right)$, can be obtained:(6)$$T_{yy}\left(y\right)=\frac{\xi T_{xx}\left(y\right)}{-2+\xi }.$$

From Equation (Equation 2) it can be obtained:(7)$$\frac{du(y)}{dy}=-\frac{Re(T_{xy}\left(y\right)-2^{-j}P_{x}y)}{\beta }.$$

Substituting Equations (Equation 6) and (Equation 7) in Equation (Equation 3) and solving the resulting equation for the tensor component $T_{xx}\left(y\right)$, it can be obtained:$$\begin{array}{c} T_{xx}\left(y\right)=-\left(\frac{\left(-1+\beta )(-2+\xi \right)}{4\epsilon ReWi(-1+\xi )}\right)(-1\pm (1-2^{3-j}\epsilon Re^{2}T_{xy}\left(y\right)Wi^{2}\;\times \\ \times \frac{\left(2^{j}T_{xy}\left(y\right)-P_{x}y\right)\left(-1+\xi \right)}{(-1+\beta )\beta }{)}^{\frac{1}{2}}). \end{array}$$

Using the hypothesis that the extra-stress tensor is zero at the channel (or pipe) centre, one solution can be discarded, resulting thus in a single solution for the tensor $T_{xx}\left(y\right)$:(8)$$\begin{array}{c} T_{xx}\left(y\right)=-\left(\frac{\left(-1+\beta )(-2+\xi \right)}{4\epsilon ReWi(-1+\xi )}\right)(-1+(1-2^{3-j}\epsilon Re^{2}T_{xy}\left(y\right)Wi^{2}\;\times \\ \times \frac{\left(2^{j}T_{xy}\left(y\right)-P_{x}y\right)\left(-1+\xi \right)}{(-1+\beta )\beta }{)}^{\frac{1}{2}}). \end{array}$$

All solutions obtained for the flow components are functions of the tensor component $T_{xy}\left(y\right)$. Therefore, it is necessary to obtain an analytical solution for $T_{xy}\left(y\right)$ to obtain analytical solutions for these components.

Substituting all the equations obtained for the fluid flow variables (Equations (Equation 6)–(Equation 8)) in Equation (Equation 4); and solving the resulting equation for the tensor component $T_{xy}\left(y\right)$, a solution for this component can be obtained for a given set of parameters Re, Wi, β, ϵ, ξ and the pressure gradient $P_{x}$.
(9)$$\begin{array}{c} T_{xy}\left(y\right)=(2^{-2j-\frac{1}{3}}(8^{j}P_{x}Re^{4}y{\left(\beta -1\right)}^{3}{\left(2\epsilon \beta \left(\xi -1\right)+\left(\beta -1\right)\left(\xi -2\right)\xi \right)}^{3}\;\\ \times \left(27\times 4^{j+1}\epsilon^{2}\beta^{2}{\left(\xi -1\right)}^{2}+9\times 2^{2j+1}\epsilon \beta \left(6\beta -1\right)\left(\xi -2\right)\xi \left(\xi -1\right)\;\right.\\ \left.+{\left(\xi -2\right)}^{2}\xi^{2}\left(9\times 4^{j}\beta \left(3\beta -1\right)-2P_{x}^{2}Re^{2}Wi^{2}y^{2}\left(\xi -2\right)\xi \right)\right)Wi^{4}\;\\ +(64^{j}P_{x}^{2}Re^{8}Wi^{8}y^{2}{\left(\beta -1\right)}^{6}{\left(2\epsilon \beta \left(\xi -1\right)+\left(\beta -1\right)\left(\xi -2\right)\xi \right)}^{6}\;\\ \times \left(27\times 4^{j+1}\epsilon^{2}\beta^{2}{\left(\xi -1\right)}^{2}+9\times 2^{2j+1}\epsilon \beta \left(6\beta -1\right)\left(\xi -2\right)\xi \left(\xi -1\right)\;\right.\\ {\left.+{\left(\xi -2\right)}^{2}\xi^{2}\left(9\times 4^{j}\beta \left(3\beta -1\right)-2P_{x}^{2}Re^{2}Wi^{2}y^{2}\left(\xi -2\right)\xi \right)\right)}^{2}\;\\ -4^{3j+1}Re^{6}Wi^{6}{\left(\beta -1\right)}^{6}{\left(2\epsilon \beta \left(\xi -1\right)+\left(\beta -1\right)\left(\xi -2\right)\xi \right)}^{6}\;\\ \times {\left(3\times 2^{2j+1}\epsilon \beta \left(\xi -1\right)+\left(\xi -2\right)\xi \left(P_{x}^{2}Re^{2}Wi^{2}\left(\xi -2\right)\xi y^{2}+3\times 4^{j}\beta \right)\right)}^{3}{)}^{\frac{1}{2}}{)}^{\frac{1}{3}})\;\\ \times \frac{1}{3Re^{2}Wi^{2}{\left(2\epsilon \beta \left(\xi -1\right)+\left(\beta -1\right)\left(\xi -2\right)\xi \right)}^{2}}+\frac{2^{1-j}P_{x}y\left(\beta -1\right)\left(\xi -2\right)\xi }{6\epsilon \beta (\xi -1)+3(\beta -1)(\xi -2)\xi }\;\\ +\left(\sqrt[{3}]{2}{\left(\beta -1\right)}^{2}\left(3\times 2^{2j+1}\epsilon \beta \left(\xi -1\right)+\left(\xi -2\right)\xi \left(P_{x}^{2}Re^{2}Wi^{2}\left(\xi -2\right)\xi y^{2}+3\times 4^{j}\beta \right)\right)\right)\\ /(3(8^{j}P_{x}Re^{4}y{\left(\beta -1\right)}^{3}{\left(2\epsilon \beta \left(\xi -1\right)+\left(\beta -1\right)\left(\xi -2\right)\xi \right)}^{3}\;\\ \times (27\times 4^{j+1}\epsilon^{2}\beta^{2}{\left(\xi -1\right)}^{2}+9\times 2^{2j+1}\epsilon \beta \left(6\beta -1\right)\left(\xi -2\right)\xi \left(\xi -1\right)\;\\ +{\left(\xi -2\right)}^{2}\xi^{2}\left(9\times 4^{j}\beta \left(3\beta -1\right)-2P_{x}^{2}Re^{2}Wi^{2}y^{2}\left(\xi -2\right)\xi \right))Wi^{4}\;\\ +(64^{j}P_{x}^{2}Re^{8}Wi^{8}y^{2}{\left(\beta -1\right)}^{6}{\left(2\epsilon \beta \left(\xi -1\right)+\left(\beta -1\right)\left(\xi -2\right)\xi \right)}^{6}\;\\ \times \left(27\times 4^{j+1}\epsilon^{2}\beta^{2}{\left(\xi -1\right)}^{2}+9\times 2^{2j+1}\epsilon \beta \left(6\beta -1\right)\left(\xi -2\right)\xi \left(\xi -1\right)\;\right.\\ {\left.+{\left(\xi -2\right)}^{2}\xi^{2}\left(9\times 4^{j}\beta \left(3\beta -1\right)-2P_{x}^{2}Re^{2}Wi^{2}y^{2}\left(\xi -2\right)\xi \right)\right)}^{2}\\ -4^{3j+1}Re^{6}Wi^{6}{\left(\beta -1\right)}^{6}{\left(2\epsilon \beta \left(\xi -1\right)+\left(\beta -1\right)\left(\xi -2\right)\xi \right)}^{6}\;\\ \times {\left(3\times 2^{2j+1}\epsilon \beta \left(\xi -1\right)+\left(\xi -2\right)\xi \left(P_{x}^{2}Re^{2}Wi^{2}\left(\xi -2\right)\xi y^{2}+3\times 4^{j}\beta \right)\right)}^{3}{)}^{\frac{1}{2}}{)}^{\frac{1}{3}}). \end{array}$$

From Equation (Equation 9), it is possible to obtain the distribution of the values of the $T_{xy}$ component analytically. After obtaining this solution, using Equations (Equation 6)–(Equation 8) one obtains the distributions for the other components of the flow, but the streamwise velocity component. This velocity component is obtained using numerical schemes by integrating Equation (Equation 7). The above variables were written as a function of *y*. For axisymmetric pipe flow, it is necessary to replace *y* with *r*. It should be emphasised that the proposed method does not require any iteration to obtain a solution for a given pressure gradient.

### 2.1. LPTT Flow versus Newtonian Flow

To compare the results obtained with the semi-analytical method for the LPTT model with the Newtonian solution, one must find the pressure gradient that gives the same flow rate for both fluid flows. The flow rate is obtained by the integral of the streamwise velocity component with respect to the wall-normal direction. The resulting flow rate should be 4/3 for channel flow and π/2 for pipe flow.

Initially, for a given initial pressure gradient in the streamwise direction $\left(P_{x}<0\right)$, it is calculated the components of the fluid flow $T_{xy}\left(y\right)$, $\frac{du(y)}{dy}$, $T_{xx}\left(y\right)$, $T_{yy}\left(y\right)$ using Equations (Equation 6)–(Equation 9) analytically; and the velocity profile u(y) is calculated by integrating Equation (Equation 7) numerically. And, with the u(y) distribution it is possible to calculate the flow rate.

An iterative Newton-Raphson’s method is adopted to find the pressure gradient ($P_{x}$) that gives the flow rate of 4/3 for channel flow and π/2 for pipe flow. The following subsection presents the verification of the semi-analytical method.

The semi-analytical method works in the following way:Set values to the parameters (β,Wi,Re,ξ,ϵ);Give an initial pressure gradient ($P_{x}$);Solve Equation (Equation 9) to find $T_{xy}$;Integrate Equation (Equation 7) to find *u*;Calculate the flow rate by solving $\int_{-1}^{1}u\left(y\right)dy$;If the flow rate is not 4/3 for channel flow or π/2 for pipe flow, an iterative Newton-Raphson’s method is adopted to give another value of pressure gradient ($P_{x}$), and we go back to step 3; otherwise, continue;Solve Equation (Equation 8) to find $T_{xx}$;Solve Equation (Equation 6) to find $T_{yy}$.

### 2.2. Verification

Here the verification of the proposed method is carried out by comparing the results obtained with the semi-analytical proposed method with two analytical solutions: the Oldroyd-B (channel flow) and the LPTT (pipe flow) models considering a purely polymeric fluid (β=0) [16]. Half of the domain was adopted in the graphics since all results are symmetric about the channel (or pipe) centre.

For the first comparison, with the analytical solution of the Oldroyd-B model, the values of the LPTT model parameters adopted were ϵ=0.0001 and ξ=0.0001. Three cases were considered for the channel flow (j=0):Case I: Re=2000,β=0.125 and Wi=1;Case II: Re=5000,β=0.5 and Wi=6;Case III: Re=8000,β=0.875 and Wi=14.

Figure 2 shows the streamwise velocity component *u* and the components of the non-Newtonian extra-stress tensor $T_{xx}$ and $T_{xy}$ distribution in the wall-normal direction *y*. It is possible to observe an excellent agreement between the results. The analytical value of the extra-stress tensor component $T_{yy}$, in this case, is zero. The results for this component obtained by the method proposed have values below $10^{-9}$, which was considered a roundoff error.

For the verification using the analytical solution of the LPTT model for purely polymeric fluid flow, as proposed in Alves et al. [16], the value of the parameter β=0.0001 was adopted in the semi-analytical method. Two cases were investigated for pipe flow (j=1):Case IV: Re=8000,ϵ=0.8,ξ=0.01 and Wi=1;Case V: Re=5000,ϵ=1.2,ξ=0.001 and Wi=4.

Figure 3 shows the streamwise velocity component *u* and the components of the extra-stress tensor $T_{xx},\;T_{xy}$ and $T_{yy}$ variation in the radial direction *r*. The comparison is carried out between the semi-analytical method results and the analytical solution proposed by Alves et al. [16]. It is also possible to observe a good agreement between the results obtained using the semi-analytical method for the LPTT model and the analytical solution for the purely polymeric LPTT model [16] in a pipe flow.

The results obtained in this section show that the proposed method could give reliable results if the fluid is modelled by Oldroyd-B (channel flow) or by a purely polymeric LPTT model (pipe flow). All the parameters had different values; therefore, this gives confidence that the proposed semi-analytical method provides the right results in channel or pipe flows. In the next section, the effects of the variation of the LPTT model parameters outside these boundaries are investigated.

## 3. Results

The present section presents the results obtained using the semi-analytical method. To explore the range and efficiency of the proposed method in this work, some values of the dimensionless parameters (Re,Wi,β,ϵ, and ξ) were investigated for channel and pipe flows.

The section was divided into four subsections. The first one is dedicated to verifying the influence of the ϵ parameter. The second one shows some results to verify the influence of the ξ parameter in the fluid flow. The third subsection explores the behaviour of the extra-stress tensor $T_{xy}$ component under certain parameter combinations; and the last subsection shows where the Valid Solution Regions for the proposed method.

### 3.1. Parameter ϵ

The influence of the ϵ parameter on the fluid flow components is analyzed here. This parameter is related to the elongational behaviour of the fluid. In the first results, presented in Figure 4, the parameters adopted for a channel flow were: Re=5000, β=0.25, ξ=0.1 and Wi=7. Five values for the ϵ parameter were used: ϵ=0.5,0.75,1.0,1.25 and 1.5. It is presented the wall-normal variation (0≤y≤1) of the streamwise velocity component *u* and the components of the extra-stress tensor $T_{xx},\;T_{xy}$ and $T_{yy}$. The value of the maximum streamwise velocity component at the middle of the channel increases with ϵ. The opposite occurs for the extra-stress tensor components; the maximum absolute values of the tensors decrease as the ϵ value increases, except for the values of the extra-stress tensor component $T_{xy}$ when the coordinate approaches the wall. For this tensor component, it is possible to observe an interesting behaviour for the fluid flow with ϵ=0.5 and 0.75; its maximum value is not at the wall as expected.

Figure 5 shows the influence of the ϵ parameter for pipe flow (j=1), adopting the following parameters: Re=6000, β=0.75, ξ=0.2 and Wi=4. The same variation adopted in the last comparisons, on the ϵ parameter, was adopted here (ϵ=0.5,0.75,1.0,1.25 and 1.5). It can be observed that the maximum streamwise velocity component *u* at the pipe center is less pronounced when the Newtonian contribution is higher (β=0.75). The maximum streamwise velocity component *u* value also increases with the value of ϵ. The same behavior of the previous comparison for the extra-stress tensor was observed here, as the value of ϵ increases, the value of the extra-stress tensor components decreases (in absolute value). For the extra-stress tensor component $T_{xr}$, the maximum value is not at the wall, and it can be noted for ϵ=0.5,0.75,1.0 and 1.25. It also can be noticed that as the Newtonian contribution (solvent contribution-β→1) increases, the magnitude of the non-Newtonian tensor components value decreases, thus making the influence of these components on the velocity profile less important.

The influence of the ϵ parameter, for pipe flow (j=1), adopting low values for the Reynolds (Re) and the Weissenberg numbers (Wi) is shown in Figure 6. The results show radial variation *r* of the streamwise velocity component *u* and the components of the extra-stress tensor $T_{xx},\;T_{xr}$ and $T_{rr}$. The adopted parameters were: Re=1, β=0.2, ξ=0.1, and Wi=0.6. The values for the ϵ parameter were (ϵ=0.1,0.2,0.3,0.4 and 0.5). It can be observed that the maximum streamwise velocity decreases as the parameter ϵ increases. This behavior is also observed on maximum values of the extra-stress tensor components (in absolute values). Figure 6 shows an opposite behavior for the maximum streamwise velocity component that the ones observed in the last two cases (Figure 4 and Figure 5).

### 3.2. Parameter ξ

To verify the influence of the ξ parameter on the LPTT model, it was generated different fluid flows by varying its value. This parameter is connected with the shear stress behavior of the non-Newtonian fluid. Figure 7 shows the wall-normal variation *y* of the streamwise velocity component *u* and the components of the extra-stress tensor $T_{xx},\;T_{xy}$ and $T_{yy}$ for a channel flow (j=0). The dimensionless parameters adopted were: Re=2000, β=0.125, ϵ=0.5 and Wi=1. Five different values for ξ were investigated: 0.01,0.05,0.1,0.15 and 0.2. It can be observed that the maximum absolute values of *u*, $T_{xx}$ and $T_{xy}$ decrease as the parameter ξ increases. The opposite occurs with the maximum absolute value of extra-stress tensor component $T_{yy}$, it increases with ξ parameter.

Figure 8 shows the wall-normal variation *y* of the streamwise velocity component *u* and the components of the extra-stress tensor $T_{xx},\;T_{xy}$ and $T_{yy}$, for a channel flow (j=0). The dimensionless numbers adopted were: Re=3000, β=0.25, ϵ=0.75 and Wi=2. The same variation adopted in the last comparisons, on the ξ parameter, was adopted here (ξ=0.01,0.05,0.1,0.15 and 0.2). The streamwise velocity component *u* and the extra-stress tensors components $T_{xx},\;T_{xy}$ and $T_{yy}$ shows the same behavior observed in the last case, for $u,\;T_{xx}$ and $T_{xy}$ their maximum absolute values decreases as the parameter ξ increase and the opposite occurs for $T_{yy}$ component as the parameter ξ increase.

Using the dimensionless parameters: Re= 10,000, β=0.5, ϵ=1.0, Wi=5 and j=0 (channel flow), the wall-normal variation *y* of the streamwise velocity component *u* and the components of the extra-stress tensor $T_{xx},\;T_{xy}$ and $T_{yy}$ are shown in Figure 9. The same variation adopted in the last comparisons, on the ξ parameter, was adopted here (ξ=0.01,0.05,0.1,0.15 and 0.2). In these results the same behavior of the one observed for the last two case was achieved, the absolute maximum value of the extra-stress tensor component $T_{yy}$ increases with the parameter ξ. The opposite occurs with the other variables.

Figure 10 shows the influence of the ξ parameter, for pipe flow (j=1), as low values of Reynolds number (Re) and the Weissenberg number (Wi) are adopted. The considered parameters are: Re=1, β=0.7, ϵ=0.01 and Wi=0.4. The ξ parameter values used here are (ξ=0.01,0.05,0.1,0.15 and 0.2). The observed behavior of the variables are the same of the last case. However, in the present case one can observe that the magnitude of the extra-stress tensor components increases substantially when low Reynolds number is adopted.

From Figure 7, Figure 8, Figure 9 and Figure 10, it is possible to observe the behavior of the variables when the ξ parameter change it value. As the parameter ξ increases, it is possible to observe that the maximum streamwise velocity component value decreases. The same behaviour can be noted on the maximum absolute value of the extra-stress tensor components $T_{xx}$ and $T_{xy}$ (or $T_{xr}$ for pipe flows) close/or at the wall. For the extra-stress tensor component $T_{yy}$ (or $T_{rr}$), the opposite behavior is observed, its maximum value increases with the dimensionless parameter ξ. This behaviour can be noted even for low values of Reynolds number, as Figure 10 shows.

It is worth mentioning that to obtain the fluid flow solution is necessary to check if the Equation (Equation 9) has a real solution. The solution for this variable needs the calculation of a square root, a cubic root, and a quotients product. The extra-stress tensor component $T_{xy}$ value is a function of the fluid flow parameters, and, for some combinations of them, the resulting value can be complex. When this happens, all the fluid flow variables are complex; therefore, this result is not the sought one.

The semi-analytical results presented here agree with the boundary conditions and solution intervals presented by Alves et al. [16]. In terms of solution ranges, the parameter ϵ can admit values within the open range $\left(0,2\right)$. For the ξ parameter, its values can be varied within the open range $\left(0,\frac{1}{2}\right)$. It is worth mentioning that, even if there is a solution for some values of these parameters beyond the above limits, many of these values do not present physical properties [20] and, therefore, they are not within the scope of this study.

### 3.3. $T_{xy}$ Behaviour

From the results presented in Section 3.1, we observe an unusual behaviour for the extra-stress tensor component $T_{xy}$, as seen in Figure 4 and Figure 5. It was observed that the maximum value of the extra-stress tensor component $T_{xy}$ does not occur at the wall with some parameter combinations. Before the semi-analytical solution was obtained, the research group used a high-order numerical simulation to obtain the laminar solution for the LPTT model in a straight channel. The solutions with both methods were compared and agreed with each other. Therefore, the behaviour of the extra-stress tensor $T_{xy}$ was double-checked. The behaviour observed here is really from the viscoelastic model and therefore is necessary to investigate which parameters influence it.

It was observed that the Reynolds number and total viscosity (either with more solvent or polymer viscosity in the mixture) do not affect the behaviour of this extra-stress component. The $T_{xy}$ component was affected by the parameters ϵ, ξ, and the Weissenberg number. An investigation was carried out using different values for these parameters and observing how the $T_{xy}$ component is affected.

For the simulations performed, the Reynolds number (Re=1000) and β (β=0.5) were kept. Figure 11 shows the variation of the ϵ parameter considering ξ=0.1 and Wi=8. The values for the parameter ϵ considered were: 0.25, 0.50, 0.75, 1.0, 1.25 and 1.5. Figure 11 shows that as the value of ϵ decreases, the maximum value of the tensor $T_{xy}$ moves towards the channel centre. This shows that a greater opposition to stretching (higher elongational viscosity) influences the maximum value for the component $T_{xy}$ to move away from the wall.

Figure 12 presents the variation in the ξ parameter values considering ϵ=0.25 and Wi=10. The values for the parameter ξ considered were: 0.01, 0.05, 0.1, 0.15, and 0.2. From Figure 12 it is possible to observe that as the value of ξ increases, the value of the tensor component $T_{xy}$ at the wall decreases. As the value of ξ increases, the maximum value of the tensor $T_{xy}$ moves towards the channel centre. This shows that the normal stress differences combined with high elongational viscosity exhibit a strong influence on this behaviour.

For our last investigation, presented in Figure 13, we performed the variation for the Weissenberg number, considering ϵ=0.25 and ξ=0.2. The values for the parameter Wi considered were: 3.0, 4.0, 5.0, 6.0, and 7.0. It is possible to observe, from Figure 13, that as the Weissenberg number increases, the maximum value of the tensor component $T_{xy}$ moves towards the channel centre.

From the analysis carried out, it was possible to verify the influence of these parameters on the behaviour of the extra-stress tensor $T_{xy}$ component. Parameter values that emphasize this behaviour were chosen for the simulations. These values comprehend low values for ϵ, ξ close to 0.2, and Weissenberg numbers higher than 1. This behaviour arises from the combination of high elongational viscosity and a high relationship between normal stress differences and high elasticity. The physical combination of these properties causes the maximum value of the extra-stress tensor $T_{xy}$ component to move towards the channel centre. The strong interaction between fluid molecules allied with high elongational viscosity and high elasticity can explain this physical behaviour.

It is worth mentioning that this behaviour happens both for the channel and the pipe, although the simulations showed here were performed only for channels.

### 3.4. Semi-Analytical Method Limits

Numerical simulations with different parameter values were performed to observe which ones allow the existence of the solution. It was verified that the Reynolds number does not influence the existence of a solution as long as Re>0. However, the other non-dimensional parameters showed an influence. To understand which type of influence and which combinations of values are necessary to obtain a valid solution, different simulations were performed, varying the parameters ϵ,ξ and β (Figure 14), and ϵ,ξ and the Weissenberg number (Wi) (Figure 15).

Adopting fixed values for Re and Wi and varying the values of the other parameters was obtained the Figure 14. The parameters interval adopted was 0≤ϵ≤2, 0≤ξ≤0.5 and 0.1≤β≤0.9. Figure 14 presents the Valid Solution Region (VSR) where it is possible to obtain the solution of the flow for different values of β. The line pointed out as β=0.1 shows the limits of a combination (ϵ,ξ) values where the solution is valid (VSR). The VSR increases with β. For smaller values of β (higher polymer viscosity in the mixture), the values of ϵ and ξ cannot be as large as, for example, the value of β being 0.9.

To obtain Figure 15, values for Re and β were kept constant. For the parameter ϵ, it was considered the interval (0,0.75). For the ξ variation, it was maintained the same variation performed for the Figure 14, and for the Weissenberg number, the values: 1,2,3,5 and 10 were considered. Figure 15 presents the regions for the limitation of obtaining the solutions. It can be observed that, for Wi=1, it is possible to obtain solutions for small values of the parameter ϵ, even for values of ξ greater than 0.2. On the other hand, as the value of Wi increases, it is possible to observe that the solution becomes more sensitive for smaller values of both ϵ and ξ. All solutions exist for parameter ϵ>0.75.

In general, the solution presented in this paper has limitations when considering low values for β. This limitation is due to the impossibility, in these cases, of considering a higher elongational viscosity for the LPTT model (low values of ϵ) and also a more significant influence of the differences in normal stresses (higher values for the parameter ξ). Therefore, in order to obtain solutions considering high elongational viscosity and also a more significant influence of normal stress differences, it is necessary that the values for the parameter β are higher than 0.3, as can be observed in Figure 14.

It is worth mentioning that the valid solution region considering channel flow was also verified for pipe flow, and the results remain the same.

## 4. Conclusions

This paper presents a semi-analytical method for the laminar steady channel and pipe flows of the LPTT fluid, with elongational and shear parameter variations. For the verification of the proposed semi-analytical method, its results were compared with the Oldroyd-B model analytical solution, and the solution presented by Alves et al. [16] for the LPTT model without solvent viscosity (β→0). The verification results obtained by the semi-analytical method proposed in this work showed a good agreement compared to both analytical solutions used as references.

The results presented explored the effect of the parameters ϵ and ξ on the velocity profile and the non-Newtonian extra-stress tensor components. From the analysis, it was possible to verify that the parameter ϵ reduces the impact of the tensor components on the velocity profile when it is increased for a high Reynolds number.

The parameter ξ has the opposite effect on the maximum value of the streamwise velocity component. As the value of this parameter increases, the velocity profile in the middle of the channel (or pipe) decreases. On the other hand, the extra-stress tensor components $T_{xx}$ and $T_{xy}$ (or $T_{xr}$ for pipe flows) decrease (in absolute value) as parameter ξ increases. For the extra-stress tensor component $T_{yy}$ (or $T_{rr}$), its absolute value increases with the parameter ξ. The solution for the simplified LPTT model, with ξ=0, for this tensor component is zero.

Another interesting behaviour was observed for the extra-stress tensor component $T_{xy}$ (or $T_{xr}$ for pipe flows). Its maximum value moves towards the channel centre with a specific combination of the parameters ϵ, ξ, and the Weissenberg number. It was observed that the combination of high elongational viscosity, the high relationship between normal stress differences, and high elasticity could be responsible for this behaviour.

It was explored for which values of the parameters are present in the flow, it is possible to obtain the solution. In other words, the limitations of the presented solution were explored.

## Figures and Tables

**Figure 1 polymers-14-04675-f001:**
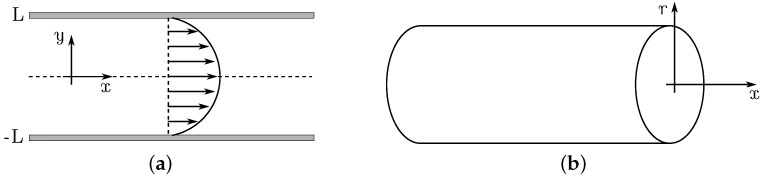
Poiseuille planar channel (**a**), and Poiseuille pipe (**b**) flows.

**Figure 2 polymers-14-04675-f002:**
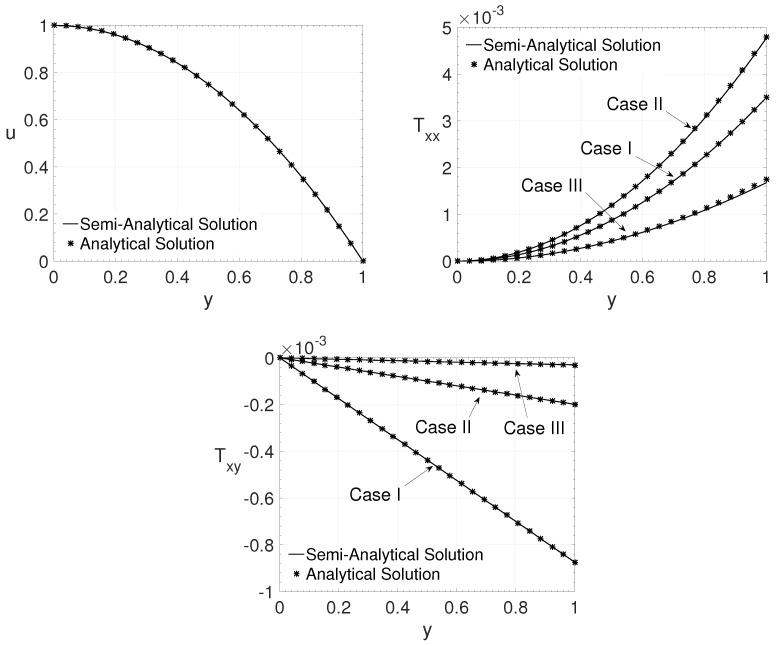
Wall-normal variation *y* of the streamwise velocity component *u* and the extra-stress tensor components $T_{xx}$ and $T_{xy}$ for the cases I, II and III.

**Figure 3 polymers-14-04675-f003:**
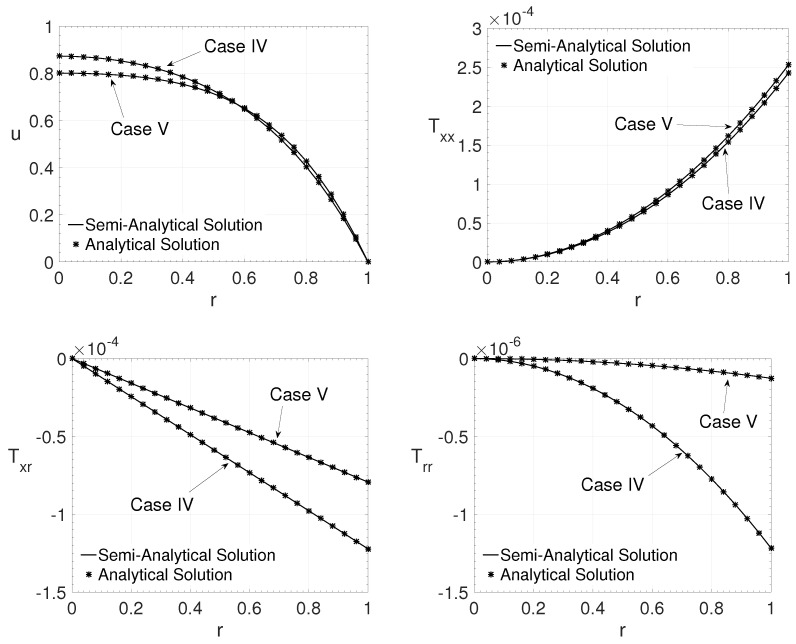
Radial variation *r* of the streamwise velocity component *u* and the extra-stress tensor components $T_{xx},\;T_{xr}$ and $T_{rr}$ for the cases IV and V.

**Figure 4 polymers-14-04675-f004:**
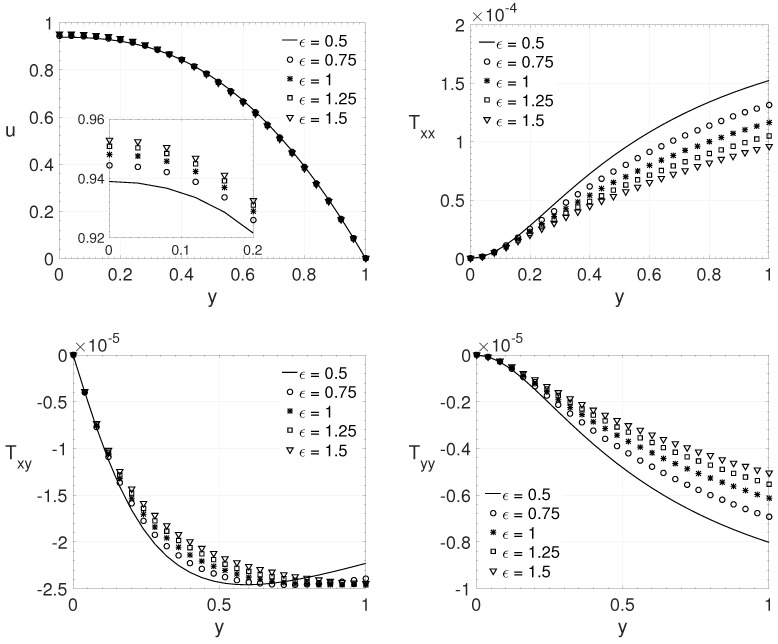
ϵ influence on the wall-normal variation *y* of the streamwise velocity component *u* and the extra-stress tensor components $T_{xx},\;T_{xy}$ and $T_{yy}$. Dimensionless parameters: Re=5000, β=0.25, ξ=0.1 and Wi=7.

**Figure 5 polymers-14-04675-f005:**
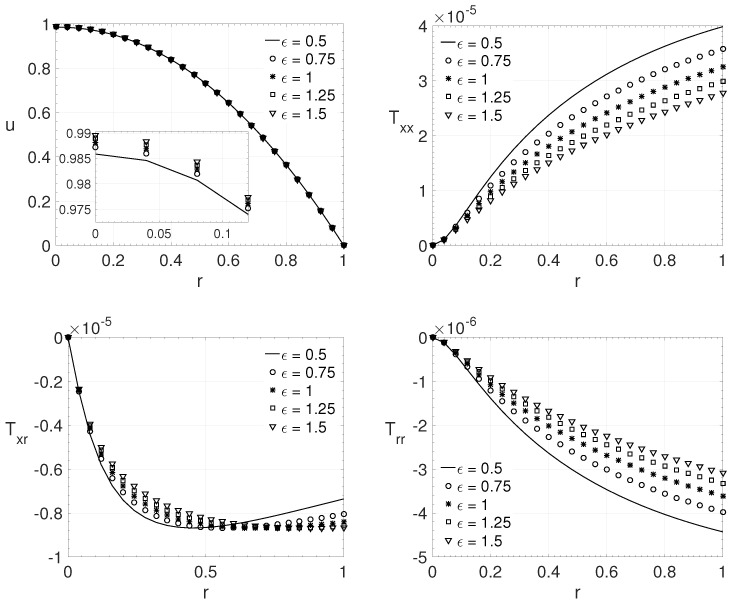
ϵ influence on the radial variation *r* of the streamwise velocity component *u* and the extra-stress tensor components $T_{xx},\;T_{xr}$ and $T_{rr}$. Dimensionless parameters: Re=6000, β=0.75, ξ=0.2 and Wi=4.

**Figure 6 polymers-14-04675-f006:**
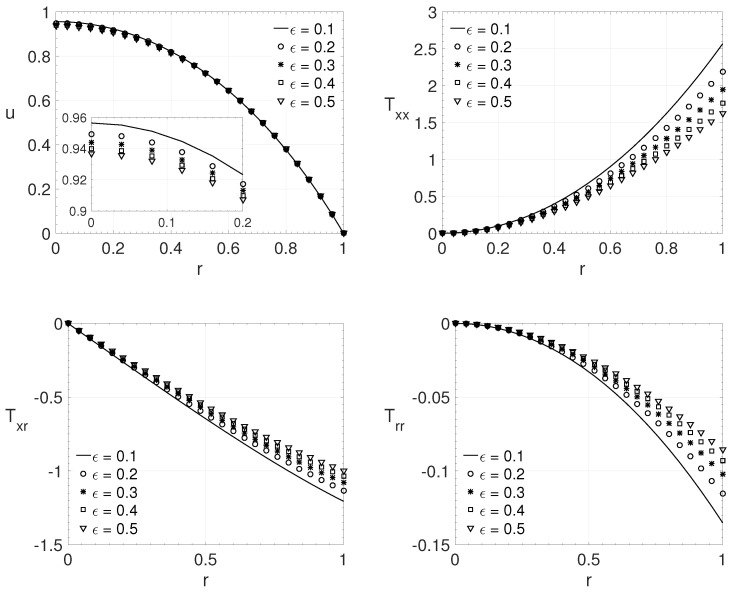
ϵ influence on the radial variation *r* of the streamwise velocity component *u* and the extra-stress tensor components $T_{xx},\;T_{xr}$ and $T_{rr}$. Dimensionless parameters: Re=1, β=0.2, ξ=0.1 and Wi=0.6.

**Figure 7 polymers-14-04675-f007:**
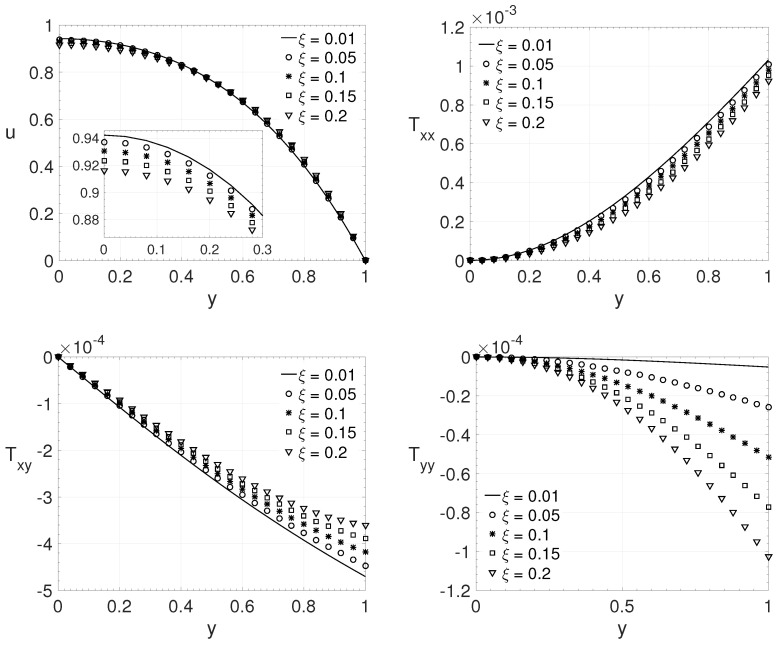
ξ influence on the wall-normal variation *y* of the streamwise velocity component *u* and the extra-stress tensor components $T_{xx},T_{xy}$ and $T_{yy}$. Dimensionless parameters: Re=2000, β=0.125, ϵ=0.5 and Wi=1.

**Figure 8 polymers-14-04675-f008:**
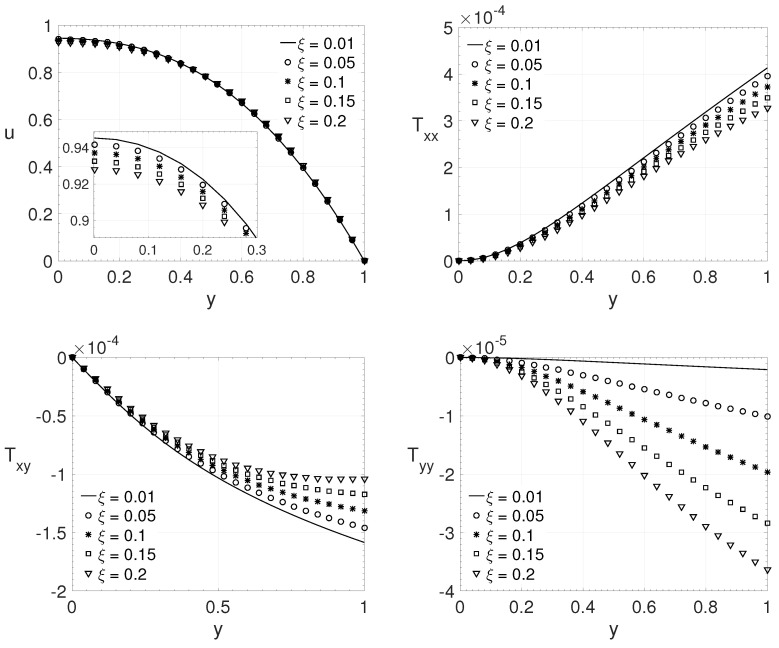
ξ influence on the wall-normal variation *y* of the streamwise velocity component *u* and the extra-stress tensor components $T_{xx},\;T_{xy}$ and $T_{yy}$. Dimensionless parameters: Re=3000, β=0.25, ϵ=0.75 and Wi=2.

**Figure 9 polymers-14-04675-f009:**
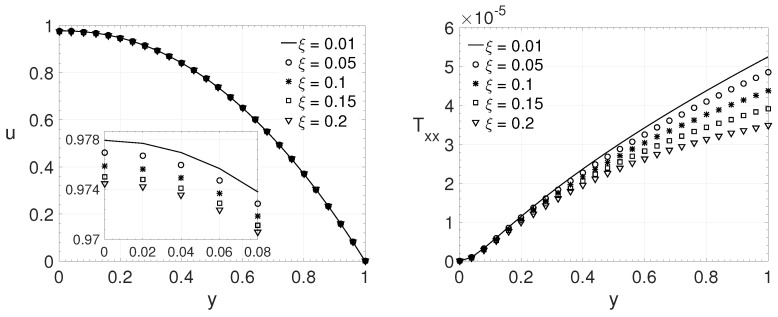
ξ influence on the wall-normal variation *y* of the streamwise velocity component *u* and the extra-stress tensor components $T_{xx},\;T_{xy}$ and $T_{yy}$. Dimensionless parameters: Re= 10,000, β=0.5, ϵ=1 and Wi=5.

**Figure 10 polymers-14-04675-f010:**
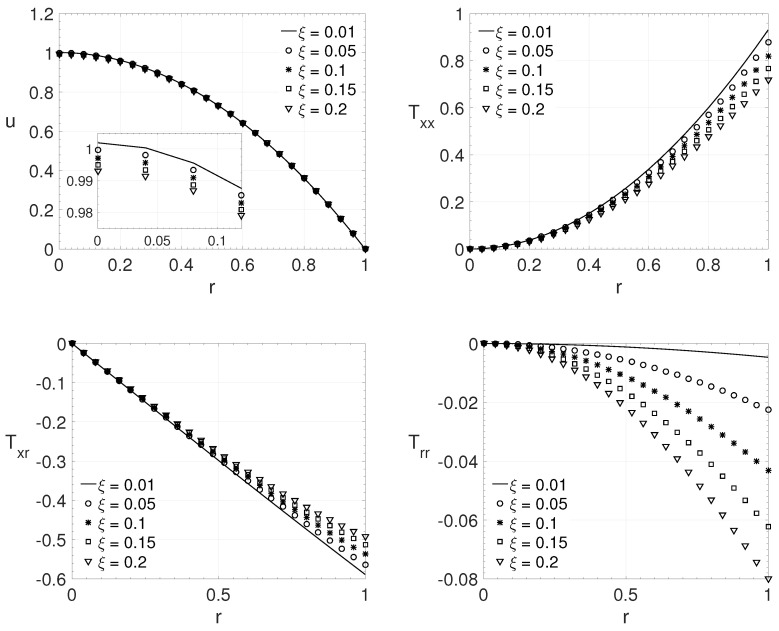
ξ influence on the radial variation *r* of the streamwise velocity component *u* and the extra-stress tensor components $T_{xx},\;T_{xr}$ and $T_{rr}$. Dimensionless parameters: Re=1, β=0.7, ϵ=0.01 and Wi=0.4.

**Figure 11 polymers-14-04675-f011:**
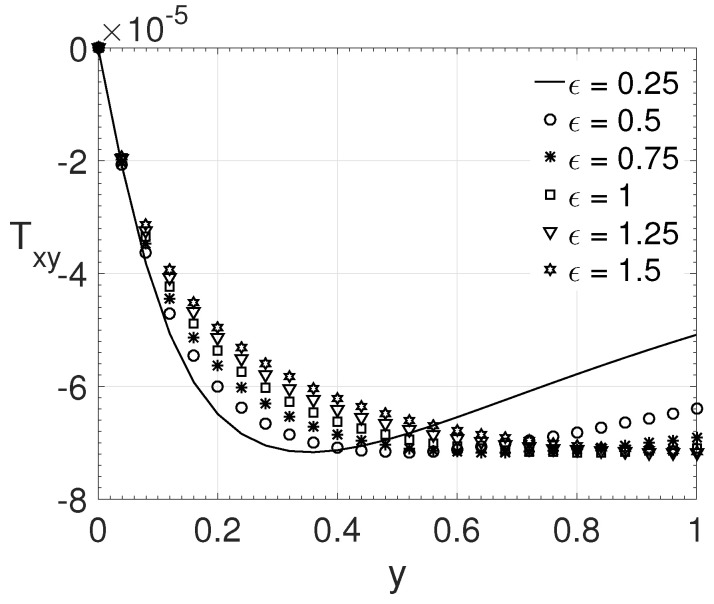
ϵ influence on the wall-normal variation *y* of the extra-stress tensor component $T_{xy}$. Dimensionless parameters: Re=1000, β=0.5, ξ=0.1 and Wi=8.

**Figure 12 polymers-14-04675-f012:**
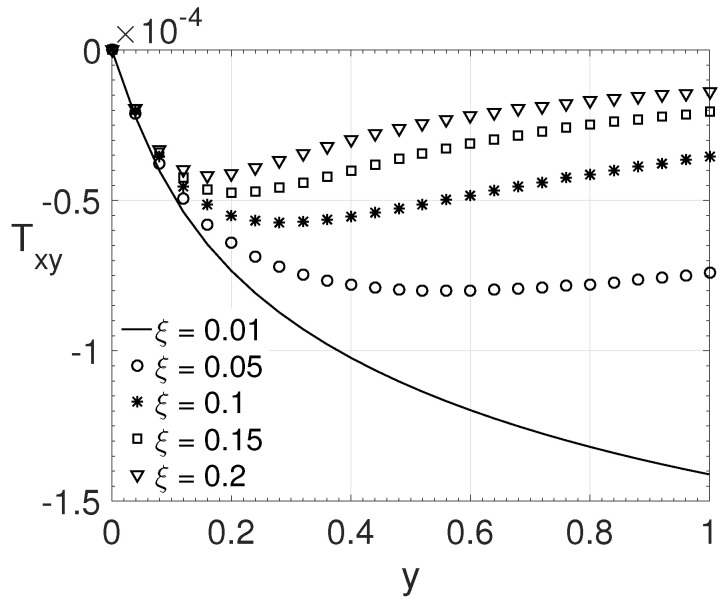
ξ influence on the wall-normal variation *y* of the extra-stress tensor component $T_{xy}$. Dimensionless parameters: Re=1000, β=0.5, ϵ=0.25 and Wi=10.

**Figure 13 polymers-14-04675-f013:**
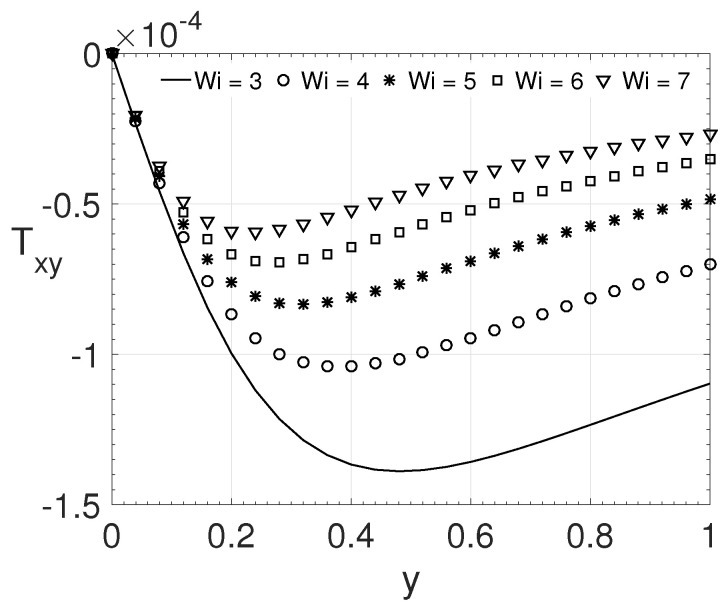
Wi influence on the wall-normal variation *y* of the extra-stress tensor component $T_{xy}$. Dimensionless parameters: Re=1000, β=0.5, ϵ=0.25 and ξ=0.2.

**Figure 14 polymers-14-04675-f014:**
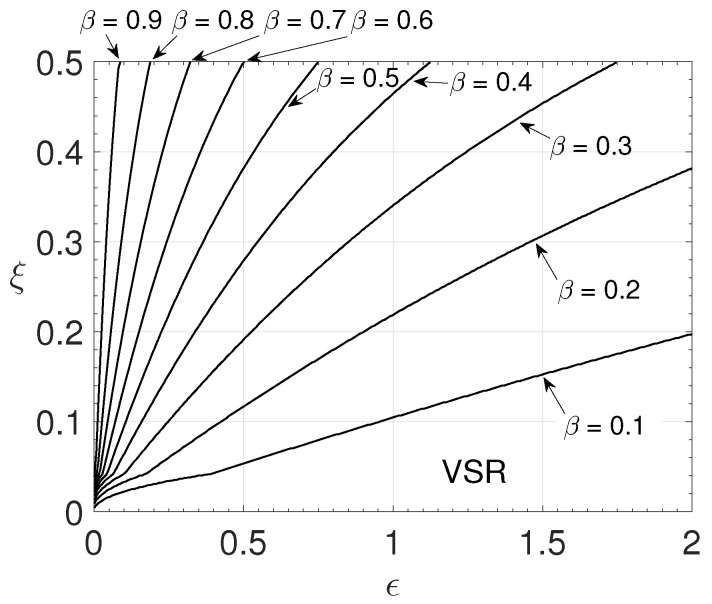
Valid solution region for different β values with ϵ and ξ-Re=1000 and Wi=3.0.

**Figure 15 polymers-14-04675-f015:**
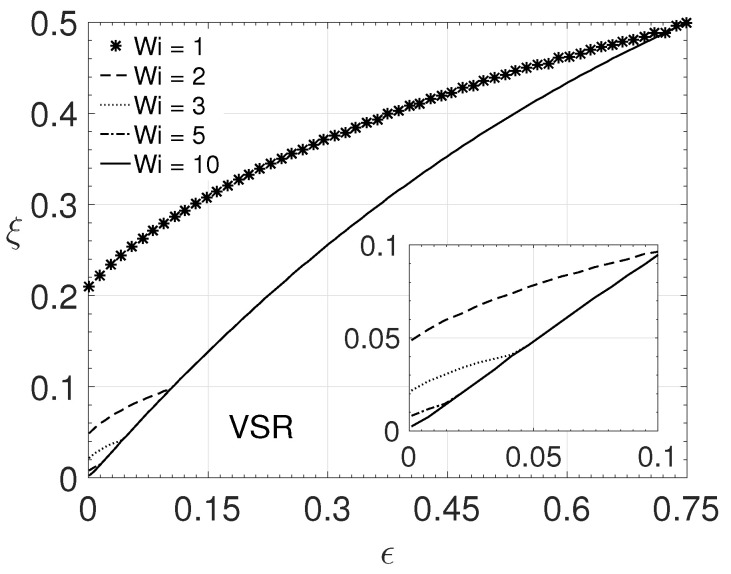
Valid solution region for different Wi values with ϵ and ξ-Re=2000 and β=0.5.

## Data Availability

Not applicable.

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
