# Peer review of "A Semi-Analytical Method for Channel and Pipe Flows for the Linear Phan-Thien-Tanner Fluid Model with a Solvent Contribution"

_polymers, 2022, doi:10.3390/polym14214675_

Round 1

Reviewer 1 Report

See the reviewer's comments file.

Minor revision is recommended for the manuscript to be published in Polymer. (1) The introduction is not written in a systematic way. The authors have written the governing equations at the start of the introduction part which is not valid. This is the part of section 1 (Mathematical formulation). So, the authors are advised to write Eq. (1) and subsequent equations in section 1. For the better presentation of the paper, the authors must read and cite the following papers. https://dx.doi.org/10.1002/zamm.202000207, https://dx.doi.org/10.1007/s13399-022-02961-9 (2) In verification, the values of Reynolds number are taken as 2000, 5000 and 8000. Why are such large values used? (3) On what grounds, the authors claim that they are using the semi-analytical method? (4) What are the assumptions for the fluid flow i.e. the flow is in-compressible, compressible etc.? (5) Include the skin friction coefficient. (6) Provide the comparison with published work. (7) Provide the nomenclature. Thanks 1/1

Reviewer 2 Report

This paper describes a semi-analytical method to obtain the laminar steady state solutions in viscoelastic channel flow and pipe flow. The authors validate their semi-analytical solutions with the analytical solutions and they agree well. The authors then use their method to perform the parameter analysis within the Linear Phan-Thien-Tanner (LPTT) constitutive equation, varying different Reynolds number, solvent viscosity contribution in the homogeneous mixture, elongational parameter, shear parameter and the Weissenberg number. This paper is of interest for readers in the Polymers and some comments below may help the authors to improve this manuscript. 

1. My major question is about the notion 'semi-analytical' method. From my current understanding, this work present a hybrid analytical and numerical method to obtain the steady state laminar solutions. As the author mentioned in page 5, they require an iterative Newton-Raphson’s method to obtain the corresponding pressure gradient. This Newton iteration based numerical method is quite standard in obtaining the steady state solutions and numerical continuation. Although the authors derived equations (2)-(9) analytically, these can be certainly embedded within  Newton's iteration. From my point of view, the main benefit of analytical solutions is to directly see the correlation between the governing parameters within the solutions, which is currently not available.  

2. Whether the author would comment that their method can be extended to turbulent state?

3. What is the main benefit of current 'semi-analytical' method compared with either purely numerical method or purely analytical method? It is suggested to highlight the motivation of using 'semi-analytical' method. 

4. For the equations (2)-(9), it is suggested to express them more concisely to improve the readability, or put some of them in the appendix.

5. A flow chart that highlight the computation method of the authors 'semi-analytical' method would also benefit the readers.
